# Antecedents and Consequences of Banking Customers’ Behavior towards Social Media: Evidence from an Emerging Economy

**DOI:** 10.3390/bs12120518

**Published:** 2022-12-17

**Authors:** Lei Zhang, Sher Akbar, Alin Mihai Tomuș, Alina Georgiana Solomon

**Affiliations:** 1School of Journalism and Communication, Wuhan University, Wuhan 430072, China; 2Department of Management Sciences, COMSATS University Islamabad, Islamabad 45550, Pakistan; 3Faculty of Law and Social Sciences, 1 Decembrie 1918 University, 510009 Alba Iulia, Romania; 4Faculty of Economics and Business Administration, Dimitrie Cantemir Christian University, 040042 Bucharest, Romania

**Keywords:** corporate social responsibility, attitude towards social media, electronic word of mouth, purchase intentions

## Abstract

Due to the aggressive competitive climate in practically every sector, modern firms in the digital age confront a variety of hurdles for survival and growth. Technology, mainly digital technology, has altered global business practices. To stay ahead of their competitors, marketers need to develop new strategies that make use of digital technology. Using more conventional forms of competition will not result in spectacular outcomes. In this respect, the rise of social media is a game-changer in marketing since it provides marketers with a strategic touchpoint to engage customers with a brand. Still, it is also important to note the customer’s attitude towards social media. Previous studies have, for the most part, ignored the connection between programs involving corporate social responsibility (CSR) and positive experiences for customers. As a result, the current study intends to evaluate the relationship between customer-related CSR activities on social media, customer attitude towards social media, and consumer behavioral outcomes, such as purchase intentions (PI) and electronic word of mouth (E-WOM). Information was collected from banking customers in a developing economy and evaluated with Smart PLS 4.0. According to the findings, customer-related corporate social responsibility activities carried out on social media have an effect on attitudes toward social media, customer behavioral outcomes, such as electronic word of mouth (E-WOM), and buying intentions. The findings also show that a consumer’s feelings toward a brand can bridge the gap between customer-related CSR, E-WOM, and purchase intentions. The current study’s outcomes can help policymakers comprehend the value of CSR practices from the standpoint of marketing, which is something that most CSR researchers overlook.

## 1. Introduction

Social media has become a popular arena for customers to engage in organizational relationship management, including corporate social responsibility, because of the interactive and open-source features it offers [1]. Corporate social responsibility has made customers think about building relationships with awareness and trust and shaping up customer attitudes towards social media, thus leading them to engage in relationships with the organization [1]. Although numerous studies suggest that corporate social responsibility (CSR) has a beneficial effect on customer sentiments, it shows that the success of CSR activities differs significantly across customers, brands, and enterprises [2]. Corporate social responsibility significantly impacts positive and negative electronic word of mouth [3]. Corporate social responsibility also has a strong connection with customer buying intentions [4]. Corporate social responsibility can lead to higher customer satisfaction [5]. Corporate social responsibility engagement was linked with a thorough investigation of consumer evaluation [6]. Scientists in the past also made strong connections with organizational performance [7], sustainability claims and perceived product quality [8], reputation [9], and organizational commitment [10]. All these studies mentioned above have overlooked building a relationship between corporate social responsibility and consumer behavioral outcomes. Recently, scientists have tried to portray links between corporate social responsibility and marketing-associated ideas [11,12]. Contemporary researchers are noticing this shifting approach to CSR [13]. As a direct consequence of this, corporate social responsibility has developed into a widespread strategy that cutting-edge organizations all over the world use to accomplish a variety of commercial goals [14,15,16].

Furthermore, the emergence of social media in today’s digital age has resulted in a paradigm change in the marketing field [17,18]. Unlike traditional media, social media is a double-edged sword since, on the one hand, it communicates marketing messages to potential audiences. Still, on the other hand, its interactive environment allows customers to interact with organizations [19,20].

Many companies are experimenting with using social media as a platform to engage customers in corporate social responsibility (CSR) initiatives, despite the fact that many organizations are making efforts to use social media for commercial goals [21,22]. Social media is being actively used by leading global businesses such as Ford Motors, Bosch, Johnson & Johnson, Pfizer, and Starbucks to promote and engage various stakeholders in their corporate social responsibility (CSR) operations [23]. Social media users interact with one another about a company’s CSR initiatives, which is characterized as customer-related CSR activity on social media [24,25]. It implies that company CSR initiatives should be trustable and that customers should be equally aware of company CSR schemes [26,27] that propose a strong linkage between customers’ social media corporate social responsibility initiatives and electronic word of mouth purchase intentions. Customers’ attitudes toward social media platforms and their determinants in terms of advertisement value have been studied by scientists [28]. Though, minimal effort was undertaken to examine the role of social media in checking the attitude of the consumer toward social media platforms where CSR activities are initiated and combined impact on electronic word of mouth and purchase intentions. Again, it depends on the CRS initiatives being trusted and awareness (e.g., share and like), plus attitude towards social media platforms that re linked with electronic word of mouth (negative and positive) and purchase intentions, which is the aim of this current research.

Recent research indicates that the number of people using social media platforms around the world has reached 4.54 billion, with an annual increase of 7%. China will have more than 1.02 billion internet users in 2022. Additionally, it was discovered that Chinese people used social media on a daily average of 1 hour and 57 min. Additionally, it is claimed that the total number of media users increased by 5.6 percent, or by 36 million, from 2021 to today. This demonstrates that the rate of acceptance of social media and the importance of social media are unrivaled when compared to traditional media, which serves as its counterpart. However, academics have not investigated how customers’ CSR-related social media initiatives can influence their attitudes and preferences toward a certain firm. This is a significant gap in the current body of knowledge. It is worth noting that customers’ good brand attitudes are a primary concern for marketers because they significantly influence customer purchasing intentions or preferences [29,30]. A rising corpus of contemporary scholars have begun to analyze customer journeys using the organization for elaborating consumer know-how; for example, research conducted by [31,32] are recent contributions to this connection [32], which explored the relationship between perceptions of customers’ brand value and brand preferences. In this regard, marketers need to be aware of the crucial touchpoints to engage the consumer in order to induce their buying intent, as well as other behavioral outcomes, such as positive word of mouth and advocacy behavior [33]. However, only a small number of research studies have concentrated on the significant touch points that could make the client’s experience with the company valuable. The significance of sustainable practices and conveying them to clients to affect their behavior has been overlooked in existing research. Similarly, the rise of social media has been hailed as a game-changer for engaging modern customers in significant ways [34,35]. Contemporary researchers and experts are increasingly interested in learning how new organizations might use social media to advertise their sustainable practices and improve meaningful customer experiences [35].

With this information in mind, the purpose of the current study is to fill in this vacuum by examining the role that consumers’ CSR-related actions play, making use of a social media platform to cultivate favorable feelings on the part of customers. The positive behaviors in terms of trust and awareness underneath customer-related CSR activities on social media lay the foundation to shape the attitude toward social media platforms and their impact on employee word of mouth (positive and negative) and purchase intentions [36]. With this as a backdrop, the current study intends to bridge this gap by investigating the role of customers’ CSR-related actions on a social media platform in instilling positive customer sentiments. Favorable emotions generated by consumer-related CSR initiatives on social media establish the groundwork for shaping a positive customer attitude toward a company. This study referred to customers’ attitudes towards social media based on brand attitude utilized as a mediator by [13]. The perception of the brand held by the consumer is an essential component in successfully influencing the actions of the client. Prior research, for example, has demonstrated that customer brand attitudes positively influence purchase intentions [26,37,38,39], customer experiences, attitudes, commitment, and purchase intentions. Previous studies have also examined the relationship between electronic word of mouth (positive and negative) and purchase intentions [40,41]. Unlike previous studies, the study aims to investigate the relationship between dimensions of corporate social responsibility (i.e., trust and awareness) and attitudes toward social media, considering its influence on electronic word of mouth and prospective purchases made in the setting of social media. In contrast with the literature written in the past, the current research analyzed such associations using traditional media [42,43]. Accordingly, the current research contends that customer-related corporate social responsibility (CSR) activities carried out on social media have a positive impact on customers’ word-of-mouth (WOM) and their intentions to make a purchase, whereas brand attitude acts as a mediator between customer-related CSR, WOM, and purchase intentions.

The current study contends that customer-related CSR dimensions (such as trust and awareness) about social media take a favorable influence on consumers’ positive and negative electronic word of mouth and intentions to purchase a particular product or to avail services, whereas customer attitude towards social media acts as a mediator. The proposed conceptual framework applied to Chinese banks is tested for several reasons, such as the humanized industry offering the same sorts of products, which leads to a question about customer retention. A ray of hope will be given to policymakers that thorough and well-planned corporate social responsibility initiatives underneath the marketing umbrella can hold customers retained and only lead to electronic word of mouth and purchase intentions. Secondly, in the presence of all sectors in the country, the banking sector offers an established and visible CSR structure, but CSR activities are still in their infancy based on social media. Due to the high volatility factor existing in the banking industry and competitiveness, this is quite challenging for every bank in comparison with the rest of the market.

Therefore, integrating CSR underneath the marketing umbrella makes the banking industry realize more about customer retention and ensures a competitive advantage through social media platforms. This study is going to add to the current literature a variety of ways with regards to corporate social responsibility and its market domain [44]. Previous studies emphasize the linkage between corporate social responsibility and organizational performance variables.

In a similar manner, the research will contribute to the existing body of CSR literature in the context of customer behavior. It proposes that customer-related corporate social responsibility (CSR) initiatives on social media produce good emotions among customers, which ultimately impact the customers’ behavior in a positive way by cultivating an attitude, EWOM (both positive and negative), and purchase intentions. In conclusion, the current study contributes to the limited body of literature on CSR in social media. This is because most studies conducted in the past have concentrated on traditional media. In the following paragraphs, you will find an explanation of the component of the literature review devoted to the formation of hypotheses and the presentation of a suggested study model.

## 2. Literature Review and Research Hypotheses

CSR-related social media activities are conducted between an organization and its customers. Businesses employ and rely heavily on various virtual media platforms, such as organizational websites, Facebook pages, Twitter content, etc., to develop a conversation with clients [45]. Customers have a more positive impression of a company that uses social media to communicate with its stakeholders about corporate social responsibility [13,46]. Customers who feel that the company treats them fairly and openly are more likely to have a favorable impression of the brand on social media [47]. CSR (customer service) communication in the digital era is a powerful marketing approach that strengthens customers’ emotional and logical connections to a company [48,49]. The idea is that companies may win over customers by letting them tag along on their social media “customer service journey” and letting them read their corporate social responsibility messages [50]. Customers can voice their opinions about the company and communicate with other users by sharing, liking, and commenting on social media posts [51,52]. Through social media’s interactive features, users form emotional connections with the company, which in turn boosts brand loyalty [53]. Customers have a favorable impression of a company because they are able to convince themselves that the company not only cares for its customers but also responds positively to society when they see an organization sharing its CSR activities on social media. This gives customers the impression that the company not only cares for its customers but also responds positively to society. Because of this, clients are left with a positive impression of the company [54]. Consequently, it is rational to claim the following hypotheses.

**H1.** 
*Customer-related CSR trust positively relates to a customer’s attitude towards social media platforms.*


**H2.** 
*Customer-related CSR awareness positively relates to a customer’s attitude towards social media platforms.*


Customers base their purchasing decisions on their organization’s health and an appraisal of the goods. If customers have a favorable view of a company’s CSR efforts, it is anticipated that their appreciation of the company will increase [5]. However, evidence indicates that clients frequently do not comprehend CSR programs due to their limited knowledge of CSR initiatives [55]. If they are not constantly considered, it is rather difficult to accomplish the CSR effect with a purchasing purpose. Customers’ comprehension of CSR initiatives will depend at least in part on the quality of CSR information supplied to them by the organization [55]. The concept of corporate social responsibility (CSR) has been around in China for the better part of two decades, but it is still in its infancy, and most consumers lack a good level of understanding and information about the value of CSR. The advent of social media is a game-changer in this regard because it gives businesses a forum in which to discuss their corporate social responsibility (CSR) strategies with various interested parties, including customers, and to draw attention to the positive effects of their CSR work in the local community and the environment [56,57]. Because of the two-way nature of communication between the firm and its clients, it is vital that the CSR activities of a business that directly affect those customers be carried out on social media. The social and engaging nature of social media is a significant factor in increasing customers’ propensity to make purchases from a certain business [58], emphasizing the role of social media interaction between organizations and customers leading to an increase in customer confidence and customers’ purchasing intentions. Prior research on consumer response and CSR has focused on behavioral response and perceived reaction, with the former relating to external response and the latter to internal response [59,60]. The perceived reaction is the internal response of customers, and it describes how CSR initiatives influence customer perceptions and how customers evaluate a business in terms of sustainable practices. Behavioral responses are the exterior responses of customers, which frequently involve the influence of CSR activities on customer behavior, such as buying behavior, loyalty, purchase intentions, and willingness to pay a premium price [61]. Customers are more likely to accept services and products from a respected company that actively engages in CSR initiatives [62]. The present study concludes that when a firm interacts with its customers on social media by sharing CSR efforts, trust is built and awareness is increased. It is predicted that customers will respond positively to such socially responsible behavior, which will ultimately influence their positive and negative electronic word of mouth, as well as their purchasing inclinations.

**H3.** 
*Customer-related CSR trust positively relates to a customer’s electronic word-of-mouth.*


**H4.** 
*Customer-related CSR trust positively relates to a customer’s purchase intentions.*


**H5.** 
*Customer-related CSR awareness positively relates to a customer’s electronic word-of-mouth.*


**H6.** 
*Customer-related CSR awareness positively relates to a customer’s purchase intentions.*


Employing a CSR program raises the profile of a company and inspires more interaction with its customers. As a result, it stands to reason that a company’s participation in CSR initiatives will generate favorable word of mouth among its customers [63]. Many consumers evaluate a company’s social responsibility alongside the quality of the products and services it offers. A company’s public profile can only benefit from its dedication to CSR [64]. In what is known as the “halo effect,” the public’s favorable impression of a company is boosted because of that business’s corporate social responsibility (CSR) initiatives. Customers are attracted to the company’s products and services thanks to this balanced view of the company through trust and awareness [65]. Customers who use a business with active CSR programs tend to have a more favorable impression of that business and are more likely to recommend its products and services to others. When clients see a company, they do business by actively participating in CSR programs, they develop a favorable impression of that business, and they are more likely to talk positively about it. This, in turn, increases the company’s positive word of mouth and, on the contrary [66], if the company does not provide what customer desires, than the trust and the awareness factor is missing, and the attitude of the customer is going to be negative, which will lead to negative electronic word of mouth and which lead to low purchase intentions [67]. When customers have a positive experience with a firm, they are more likely to spread the word about what happened to be concluded in a longitudinal study [68]. E-WOM is defined as “word-of-mouth” communication that is spread among potential clients by means of electronic media [57]. In various services-related sectors, electronic word of mouth, i.e., positive and negative, can lead to high and low service utilization by customers [69].

Researchers today agree that electronic word of mouth (EWOM) is crucial for expanding a business’s customers [70]. Scientists have created associations between positive electronic word of mouth and brand preferences [71], purchase intentions [72], and loyalty [73]. Many argued the importance of negative electronic word of mouth with purchase intentions [74], brand loyalty [75], brand switching behavior [76], and corporate social responsibility [77]. Therefore, WOM is crucial for businesses that successfully employ it to boost sales or promotion. Thanks to the internet and other forms of online media, word of mouth (WOM) communication has become increasingly commonplace. Furthermore, businesses engage in CSR for a variety of reasons, such as the word of mouth (WOM) benefits that can be gained [27]. Participating customers are more likely to brag about the company’s CSR initiatives to their circle of acquaintances. By engaging in CSR, businesses can enhance their reputation as caring, ethical entities in the marketplace [78]. Companies in the modern day use social media to explain their CSR initiatives to their clientele and other interested parties. Customers who are proud to be associated with a company will share positive feedback about that company with their online communities after witnessing such actions [79]. However, things become ugly for the company if social media users share negative word of mouth, which can spread up to a mass effect on purchase intentions [80].

**H7.** 
*A customer’s attitude towards social media positively relates to a customer’s electronic word of mouth.*


**H8.** 
*A customer’s attitude towards social media positively relates to a customer’s purchase intentions.*


Long-standing studies have shown that when customers learn that a company cares about the community, their perception of the brand improves [47,81]. This is because when customers learn that a company cares about the community, they form favorable mental associations with the brand. When consumers learn that a company cares about the community, they form favorable impressions of that company and its products. The present study hypothesized, in line with the theory of reasoned action, that brand attitude influences both electronic word of mouth (E-WOM) and consumers’ propensity to make purchases [82]. The advent of social media has given customers a more participatory and adaptable venue for airing their grievances with a company and spreading viral messages than traditional media [83]. According to the theory of reasoned action, customer brand attitude is positively correlated with purchase intention. In the same vein, it can be expected that the CSR activities of an organization will influence customers’ brand attitudes, which will, in turn, increase customers’ purchasing intentions for a brand or organization [27,84]. Recent studies in the scholarly literature have explored how factors such as openness, trust, and social responsibility might be used to further develop the rational choice theory of reasoned action. The authors of [85,86] proposed a model in which consumers’ perceptions of an organization’s commitment to social responsibility and transparency drove both their positive feelings toward the brand and their propensity to engage in E-WOM and make purchases. Please refer to Figure 1 for research framework.

## 3. Research Methodology

The authors chose these banks because they are leaders in corporate social responsibility (CSR) and have a strong social media presence dedicated to CSR. The Bank of China, the Agriculture Bank of China, the China Construction Bank, Hubei Bank, and the Industrial and Commercial Bank of China were the five banks chosen after an initial evaluation by the authors. Although they have a global presence, these banks use social media extensively to communicate with Chinese citizens. So, from the point of view of the current investigation, picking these banks makes sense. The information was acquired from people in Hubei province, one of the largest provinces in China, who were using these banks to manage their personal finances. Additionally, these banks have numerous outposts spread around the above-mentioned locale. The participants in the present study were approached by the authors when they were leaving a specific bank branch or waiting in line for an automated teller machine (ATM). The data were collected from the account holders the locale of Hubei province. The majority of the respondents were approached when they were leaving the bank facility. Data were collected based on prior research conducted by [13,87,88]. The respondents’ informed consent to engage voluntarily in the survey was obtained by the respondents prior to the data collection phase being activated. The responders were also informed that the confidentiality of their data would be strictly upheld and kept to themselves. The respondents also explained the significance of the current study to the survey participants and urged them to respond honestly to the questions. The authors distributed 500 questions and 304 respondents responded correctly, and the response rate was 60 percent.

The authors adopted several measures to lessen the problem of social desirability bias, such as rigorously cross-checking the survey items by subject-matter experts to ensure there was no ambiguity and that the instrument was appropriate for the present study’s goals. In the same way, the survey items were dispersed at random across the questionnaire to disrupt any predetermined order that respondents might have constructed. In a similar vein, the authors stayed present during the data collection phase to address any potential respondent challenges with completing the questions. Since the present study’s data were acquired from the same respondent across time, the authors subsequently addressed the possible problem of common method bias (CMB). Accordingly, CMB is very likely; the authors tackled this question using the advice of [89,90,91], by loading all the study items into a single factor to determine if there is a single dominant component. Most of the variance was explained by a single factor of 32.6%; the findings of single factor analysis showed that there is no such dominant factor, suggesting that CMB is not a potential issue in the data of the present study. Table 1 displays the respondents’ demographic information.

### Measurement of Variables

CSR is an adapted scale with four items. A total of six questions, three from CSR awareness and equally three from CSR trust, one each from both constructs, have been dropped in the pilot testing due to low factor loadings in the further analysis. Moreover, the six items to measure attitude towards social media were taken from [92,93], out of which two items were dropped in pilot testing and low factor loadings. Employee word of mouth was measured by using [94]. To measure purchase intentions, four items were taken from [95,96]. All items used in the questionnaire to measure each construct were on a 1 to 7 scale (Likert scale).

## 4. Results

Table 1 clearly depicts the demographic profile of the respondents. It implies that 70% of the respondents were male and 30% respondents were female. The majority of the respondents were from the age bracket of 20 years to 25 years. The education profile of the respondents clearly show that almost 54.3% of the respondents were a graduate, whereas 29% had a Masters and 10% possessed a PhD. Table 2 explain the descriptive statistics with a sample size of (*n* = 304), with mean values and standard deviations of (M = 5.1, SD = 1.19) indicating that average respondents agreed that they have corporate social responsibility trust as well as mean values and standard deviations of (M = 5.3, SD = 1.16), agreeing that they have corporate social responsibility awareness. Attitude toward social media has a mean value of 5.4 and the standard deviation is 1.04. Purchase intentions have a higher consistency Cronbach alpha value of 0.83, which shows greater reliability.

Authors should discuss the results and how they can be interpreted from the perspective of previous studies and of the working hypotheses. The findings and their implications should be discussed in the broadest context possible. Future research directions may also be highlighted.

Table 3 explain the Kaiser–Meyer–Olkin. The Kaiser–Meyer–Olkin measure of sampling adequacy value is 0.873, which is significant at five percent. All components based on the studied variables were reported in the rotated component matrix table (Table 4) through the applied mechanism, which is known as principal component analysis in SPSS. The rotated component matrix shows the components of all constructs, i.e., corporate social responsibility trust, corporate social responsibility awareness, attitude towards social media, electronic word of mouth, and purchase intentions. Values extracted through KMO, i.e., 0.873 and Bartlett’s test, are significant.

Table 5 clearly implies bivariate correlations of the studied variable. There is no sign of multicollinearity, as no independent variables are correlating with each other. The corporate social responsibility awareness measure showed a weak to moderate relationship with corporate social responsibility awareness. Corporate social responsibility trust (CRT) and corporate social responsibility awareness (CRA) show a weak to moderate relationship with attitude towards social media (ATS), (0.443 ** < γ < 0.446 **). Electronic word of mouth (EWM) has weak to moderate relationships with CRT and CRA with correlation values of (0.427 ** < γ < 0.319 **). Electronic word of mouth (EWM) has a weak to moderate correlation with ATS with correlation values of (0.382 **). All *p*-values less than 0.01 show a high significance.

### 4.1. Measurement of Model

The measurement model designed in Smart PLS version 4.0 based on the data from 304 respondents clearly depicts the external loadings, average variance extracted, internal consistency, and construct validity, respectively. Please refer to Figure 2 regarding measurement model estimate with beta values.

Table 6 clearly explained the factor loadings of each indicator with the justification of the cut-off point. The values of substantial item loadings between 0.40 to 0.70 are the benchmark. Hence, the present study’s external loadings are at a greater than cut-off point than the one provided by the scientist. Similarly, the criterion of item reliability representing each construct is met.

The cross loadings criterion along with the heterotrait-monotrait ratio (HTMT) is displayed in Table 7 and Table 8, respectively. It was indicated that all indicators met these criteria and that the indicators loaded the highest on their associated constructs ([97]). All values of HTMT are less than 0.85 [98], showing the discriminant validity.

### 4.2. Structural Model

In order to yield the β-values between the studied constructs, t values, *p* values, R-square, Mediating effect, Pls Predict through smart pls software, structural model is deduced on the basis of prior research conducted by [99]. The method of boot strapping was adopted. Please refer to Figure 3 for path coefficient with *p*-values and Figure 4 with path coefficient with t-values. 

### 4.3. Mediation Effect

More importantly, the assessment of variance described the endogenous latent variables (i.e., electronic word of mouth and purchase intentions). The R-square value indicates the percentage of variation in the dependent factor that could be justified by one or more predictor variables [99]. In PLS-SEM, 0.60 can be considered a substantial and acceptable value for R2, with 0.33 as mild and 0.19 as low [100]. Thus, the R2 value found for this work was 0.24, which was above the benchmark.

## 5. Conclusions, Implications, and Future Research

### 5.1. Conclusions and Implications

This study found that corporate social responsibility trust and corporate social responsibility awareness are statistically significantly related to bank customers’ electronic word of mouth and purchase intentions. According to Table 9, yielded results justify the combined effect along with T-statistics. This implies that attitude towards social media mediates the relationship between corporate social responsibility awareness and electronic word of mouth and purchase intentions in the banking sector of China. Moreover, the results of the current study revealed that attitude towards social media mediates between corporate social responsibility trust and purchase intentions and electronic word of mouth in the banking sector of China. Therefore, it means that the more the banks create trust and awareness among customers, the better the electronic word of mouth and purchase intentions. The same is true for the mediator, which is customer attitude towards social media. If banks ensure trust and awareness underneath the corporate social responsibility umbrella, then electronic word of mouth and purchase intentions will be higher. If we acquire more data, then the results will be improved. These findings are consistent with the research conducted by [13,67], which reveals that trust and awareness are positively related to electronic word of mouth and purchase intentions, with the mediation effect of attitude. Furthermore, the results revealed that attitude towards social media mediates between trust, awareness, and employee word of mouth and purchase intentions.

This will help banks to work more through the creation of trust and awareness underneath the corporate social responsibility umbrella with the formation of positive attitudes of customers for social media, thus leading to improvement of electronic word of mouth and purchase intentions helping them to gain a competitive edge in the banking industry. Future research should focus on the negative aspects of electronic word of mouth, as well as the purchasing intents of social media users in their respective industries. The new investigation has significant ramifications for academics and reactionaries. First, the research focuses on banking clients via the prism of corporate social responsibility. The present study also contributes to the existing literature on sustainable practices in underdeveloped nations. The final theoretical implication of the present study is the addition of social media attitude as a mediator in the suggested model. This study contributes to the current body of information by highlighting the role that corporate social responsibility (CSR) activities implemented via social media can have in modifying customer mindsets and, consequently, impacting word of mouth (WOM) and purchase intentions.

### 5.2. Future Research

For future studies, this current model should be applied to longitudinal studies in other industry settings. The perspective of using various social media platforms from customers, as well as organization ends, can also lead to interesting findings. Furthermore, electronic word of mouth can be tested with purchase intentions to lead us to finding out how engaged a customer is towards the services. This model paves the way for future academics to apply other dependent variables, such as dimensions of electronic word of mouth and customer engagement.

## Figures and Tables

**Figure 1 behavsci-12-00518-f001:**
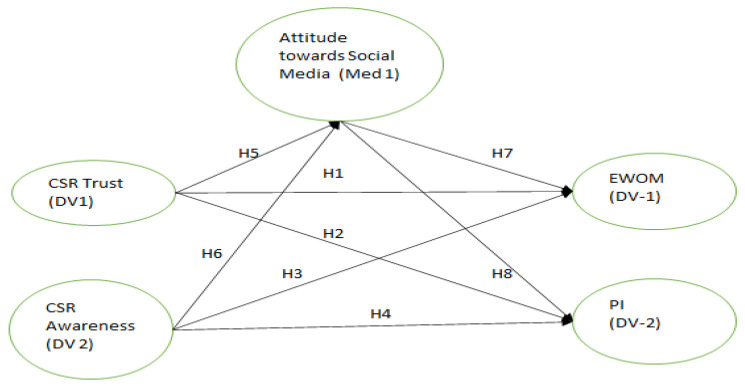
Research framework.

**Figure 2 behavsci-12-00518-f002:**
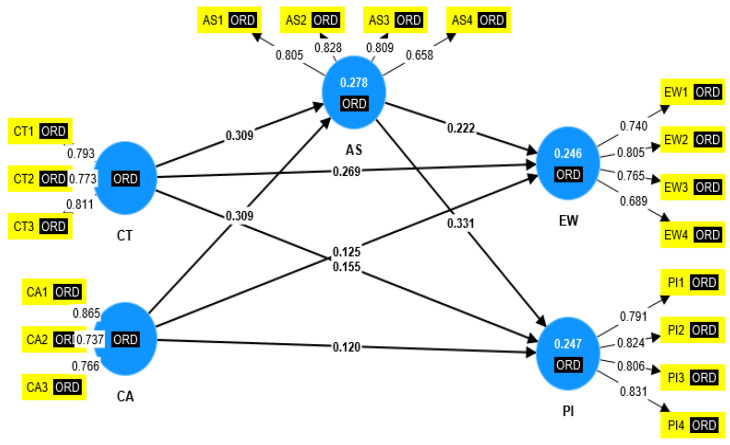
The measurement model estimate with beta values.

**Figure 3 behavsci-12-00518-f003:**
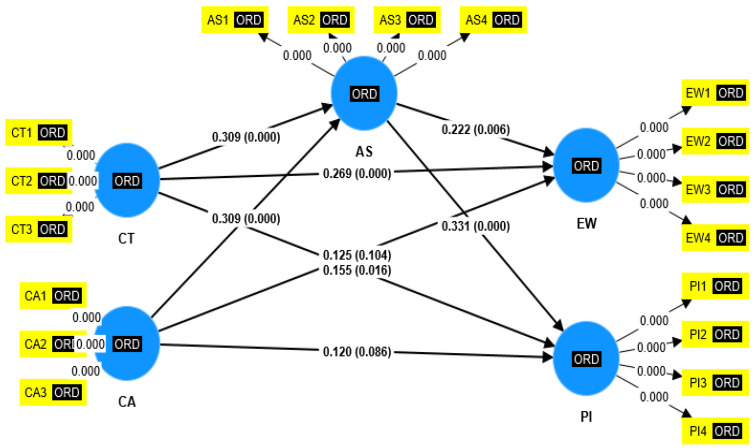
Path coefficient with *p*-values.

**Figure 4 behavsci-12-00518-f004:**
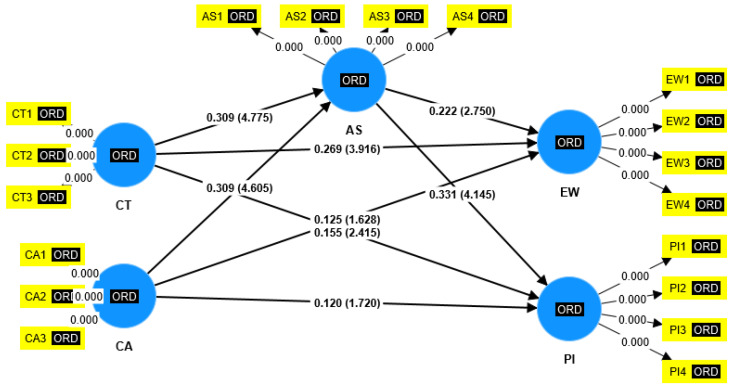
Path coefficient with t-values.

**Table 1 behavsci-12-00518-t001:** Demographic profile of the respondents.

Demographics	Frequency	Percentage (%)
**Gender**		
Male	213	70
Female	91	30
**Age**		
20 yrs. to 25 yrs.	88	28.9
26 yrs. to 30 yrs.	56	18.4
31 yrs. to 35 yrs.	80	26.3
36 yrs. to 40 yrs.	40	13.2
41 yrs. to 45 yrs.	25	8.2
46 yrs. to 50 yrs.	10	3.3
50 yrs. and above	5	1.6
**Education**		
Graduate	165	54.3
Masters	86	28.9
PhD	30	9.9
Other	23	7.6

Source: author’s own.

**Table 2 behavsci-12-00518-t002:** Descriptive statistics.

Variables	N	Mean	Std. Dev	Skewness	Kurtosis	Cronbach Alpha	No. of Items
**CRT**	304	5.1	1.19	−0.40	−0.13	0.71	03
**CRA**	304	5.3	1.16	−0.96	0.84	0.70	03
**ATS**	304	5.4	1.04	−0.98	1.58	0.78	04
**EWOM**	304	5.8	0.91	−0.85	0.399	0.74	04
**PI**	304	6.0	0.94	−1.3	1.99	0.83	04

**Table 3 behavsci-12-00518-t003:** KMO and Bartlett’s test.

The Kaiser–Meyer–Olkin Measure of Sampling Adequacy		0.873
Bartlett’s test of sphericity	Approx. chi-square	1930.191
	Df	153
	Sig	0.000

**Table 4 behavsci-12-00518-t004:** The rotated component matrix.

The Rotated Component Matrix					
Items	Component				
	1	2	3	4	5
CR1					0.728
CR2					0.805
CR3					0.645
CR4				0.677	
CR5				0.790	
CR6				0.759	
AS1		0.667			
AS2		0.769			
AS3		0.776			
AS4		0.661			
EW1			0.562		
EW2			0.738		
EW3			0.779		
EW4			0.668		
PI1	0.777				
PI2	0.779				
PI3	0.747				
PI4	0.728				
Extraction method: principal component analysis					

**Table 5 behavsci-12-00518-t005:** Bivariate correlations.

CRT		CRA	ATS	EWM	PI
**CRT**	1				
**CRA**	0.443 **	1			
**ATS**	0.443 **	0.446 **	1		
**EWM**	0.427 **	0.319 **	0.382 **	1	
**PI**	0.351 **	0.312 **	0.443 **	0.515 **	1

** Significant at 0.01.

**Table 6 behavsci-12-00518-t006:** Standardized loadings, validity, and reliability of the measurement model.

Constructs/Indicators	Outer Loadings	Cronbach Alpha Values	Composite Reliability	Average Variance Extracted
CSR Trust				
CT1	0.8			
CT2	0.77	0.7	0.71	0.628
CT3	0.81			
CSR Awareness				
CA1	0.87			
CA2	0.74	0.71	0.76	0.626
CA3	0.77			
Attitude towards Social Media				
AS1	0.81			
AS2	0.83	0.78	0.79	0.605
AS3	0.81			
AS4	0.7			
Electronic Word of Mouth				
EW1	0.74			
EW2	0.81	0.74	0.74	0.564
EW3	0.77			
EW4	0.7			
Purchase Intentions				
PI1	0.72			
PI2	0.83	0.83	0.84	0.601
PI3	0.81			
PI4	0.83			

**Table 7 behavsci-12-00518-t007:** Cross loadings.

	AS	CA	CT	EW	PI
AS1	0.805	0.427	0.340	0.344	0.412
AS2	0.828	0.314	0.413	0.368	0.346
AS3	0.809	0.334	0.319	0.301	0.396
AS4	0.658	0.320	0.333	0.206	0.237
CA1	0.393	0.865	0.446	0.386	0.381
CA2	0.319	0.737	0.301	0.205	0.136
CA3	0.351	0.766	0.304	0.183	0.232
CT1	0.295	0.359	0.793	0.409	0.282
CT2	0.326	0.309	0.773	0.307	0.216
CT3	0.439	0.407	0.811	0.297	0.342
EW1	0.349	0.265	0.318	0.740	0.444
EW2	0.296	0.277	0.313	0.805	0.449
EW3	0.236	0.235	0.318	0.765	0.380
EW4	0.306	0.260	0.326	0.689	0.277
PI1	0.337	0.236	0.282	0.382	0.791
PI2	0.404	0.249	0.303	0.376	0.824
PI3	0.318	0.290	0.245	0.476	0.806
PI4	0.406	0.325	0.327	0.452	0.831

**Table 8 behavsci-12-00518-t008:** Heterotrait-monotrait ratio (HTMT) correlation.

	AS	CA	CT	EW	PI
AS					
CA	0.598				
CT	0.602	0.617			
EW	0.511	0.446	0.588		
PI	0.551	0.406	0.458	0.659	

**Table 9 behavsci-12-00518-t009:** Specific indirect effect (mediation), mean, STDEV, T-values, and *p*-values.

Effect	Original Sample (O)	Sample Mean (M)	Standard Deviation (STDEV)	T-Statistics (|O/STDEV|)	*p*-Values	Decision
CA -> AS -> EW	0.068	0.071	0.032	2.121	0.034	Reject H0
CA -> AS -> PI	0.102	0.101	0.030	3.396	0.001	Reject H0
CT -> AS -> PI	0.102	0.103	0.036	2.839	0.005	Reject H0
CT -> AS -> EW	0.069	0.070	0.029	2.405	0.016	Reject H0

## Data Availability

The data will be made available upon request from the corresponding author.

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
