# Peer review of "Antecedents and Consequences of Banking Customers’ Behavior towards Social Media: Evidence from an Emerging Economy"

_behavsci, 2022, doi:10.3390/bs12120518_

Round 1

Reviewer 1 Report

a.       Respondents are only in China, in the introduction, the author can add digital reports in China, especially the use of social media and social media trends.

b.       Recent research carried out by SIMON KEMP (2020) indicates that the number of people using social media platforms around the world has reached 4.54 billion, with an annual increase of 7%, please update to report 2022

c.       In Table 4.1. The author can add the total sample and percentage at the bottom of the table. In Age : you can make simple category : 20-25 Years, ..

d.       Please add source of table : at the bottom of the table

e.       Does this research ask about the type of social media used by the company to communicate its CSR, if so, please indicate!

f.        In latent contruct, such as CT 1, CA 1 , AS1, author can replace with content of questionnaire, so that readers understand it better

g.       In section 4.2 The Author can show Results of hypotheses testing direct effect in table

h.       In 5. Conclusions, Findings, and Limitations, author can divide two section (1) Discussion, findings  and implication. In discussion, author more explain statistic result in section 4.  (direct and indirect effect) And connect the result research to the practical and Theoritical implications of the present study for the banking sector. (2) Conclussion, limitation and Future research direction.

i.         why didn't the author test the hypothesis of ewom influence on PI?

j.         Correct  References as per MDPI Reference List and Citation Style Guide

Author Response

1. Respondents are only in China, in the introduction, the author can add digital reports in China, especially the use of social media and social media trends.

Response: As per the reviewer’s comments, the latest record of the year 2022 has been mentioned from line 97 to line 102. It is updated. Changes are highlighted with yellow color for your convenience.  

2. Recent research carried out by SIMON KEMP (2020) indicates that the number of people using social media platforms around the world has reached 4.54 billion, with an annual increase of 7%, please update to report 2022

Response: As per the reviewer’s comments, the latest record of the year 2022 has been mentioned from line 97 to line 102. It is updated.

3. In Table 4.1. The author can add the total sample and percentage at the bottom of the table. In Age : you can make simple category : 20-25 Years, ..

Response: The total sample size along with the percentage response rate was mentioned on lines 364 to line 367. Age is already categorized in table 1. Changes are highlighted with yellow color for your convenience.  

4. Please add source of table : at the bottom of the table

Response: As per the Reviewer’s comments regarding the source of the table, it is incorporated underneath/at bottom of the table 1.

5. Does this research ask about the type of social media used by the company to communicate its CSR, if so, please indicate!

Response: The current research study aims to customer-related corporate social responsibility activities carried out on social media to influence attitudes toward social media, customer behavioral outcomes like electronic word of mouth (E-WOM), and buying intentions. The findings also shown that a consumer's feelings toward a brand can bridge the gap between customer-related CSR, E-WOM, and purchase intentions. In future directions it is mentioned that social media preferences opted by the organizations and used by the customers along with the comparison should not be neglected. 

6. In latent contruct, such as CT 1, CA 1 , AS1, author can replace with content of questionnaire, so that readers understand it better

Response: With reference to the reviewer’s comments, the questionnaire will be incorporated in the appendix as per journal format. Although we have also mentioned the measures from lines 387 to 393. 

7. In section 4.2 The Author can show Results of hypotheses testing direct effect in table

Response: With reference to the respectable reviewer’s comments, in table 9, T statitsics, P values, total effect along with hypothesis acceptance or rejection is mentioned from line 496 to 497.

8. In 5. Conclusions, Findings, and Limitations, author can divide two section (1) Discussion, findings  and implication. In discussion, author more explain statistic result in section 4.  (direct and indirect effect) And connect the result research to the practical and Theoritical implications of the present study for the banking sector. (2) Conclussion, limitation and Future research direction.

Response: With reference to our respectable reviewer’s comments, the sector is further divided with elaboration from line 499 to line 539.

9. Why didn't the author test the hypothesis of ewom influence on PI?

Response: The areas of the current research were corporate social responsibility awareness, corporate social responsible trust, attitude towards social media, electronic word of mouth, and purchase intentions. The aim was to find out direct and indirect associations among the mentioned variables. In future research on line 538 and line 539.  

10. Correct  References as per MDPI Reference List and Citation Style Guide 

Response: All citations and references list as per MDPI and corrected accordingly.

Reviewer 2 Report

rephrase: The model developed for this study to the test in the banking industry of China

remove: This section may be divided by subheadings. It should provide a concise and precise description of the experimental results, their interpretation, as well as the experimental conclusions that can be drawn.

Try to be more concise in the introduction.

The discussion section should be longer.

Author Response

rephrase: The model developed for this study to the test in the banking industry of China

Response: The line is removed as per the respectable reviewer’s comments from line 337 and line 338.

remove: This section may be divided by subheadings. It should provide a concise and precise description of the experimental results, their interpretation, as well as the experimental conclusions that can be drawn.

Response: It has been removed from line 364 to line 367 as per the respectable reviewer’s comments.

Try to be more concise in the introduction.

Response: Thank you, No we updated introduction section as per reviewer suggestion. Changes are highlighted with yellow color for your convenience.

The discussion section should be longer.

Response: With reference to our respectable reviewer’s comments, the sector is further divided with elaboration from line 499 to line 540.